# Insecticidal Effect of Diatomaceous Earth Formulations for the Control of a Wide Range of Stored-Product Beetle Species

**DOI:** 10.3390/insects14070656

**Published:** 2023-07-22

**Authors:** Paraskevi Agrafioti, Mariastela Vrontaki, Marianna Rigopoulou, Evagelia Lampiri, Katerina Grigoriadou, Philippos M. Ioannidis, Christos I. Rumbos, Christos G. Athanassiou

**Affiliations:** 1Laboratory of Entomology and Agricultural Zoology, Department of Agriculture, Crop Production and Rural Environment, University of Thessaly, Phytokou Str., 38446 Nea Ionia, Greece; mariastelaav@gmail.com (M.V.); marian.rigopoulou@gmail.com (M.R.); elampiri@uth.gr (E.L.); crumbos@uth.gr (C.I.R.); athanassiou@uth.gr (C.G.A.); 2Hellenic Feedstuff Industries S.A., 59300 Plati Imathias, Greece; grigokat@outlook.com (K.G.); filioan@otenet.gr (P.M.I.); 3Institute of Plant Breeding and Genetic Resources, Hellenic Agricultural Organization—DIMITRA, 57001 Thessaloniki, Greece

**Keywords:** diatomaceous earth, dust formulations, mortality, stored-product insects, inert materials

## Abstract

**Simple Summary:**

Inert dusts are promising alternatives to conventional insecticides against numerous stored-product insect species. Diatomaceous earth formulations are based on natural substances and are registered for direct application on grain in many parts of the world. The objective of this study was to evaluate the effectiveness of three diatomaceous earth formulations (namely Silicid, Celatom^®^ MN-23, and SilicoSec^®^) against adults of a wide range of stored-product beetle species. Specifically, seven stored-grain beetle species were tested, including three primary colonizers and four secondary colonizers, which are commonly found in stored cereals and other relevant commodities in Greece. The experimental units for the bioassays were plastic cylindrical vials. Twenty grams of soft wheat was filled in each vial and twenty adults of each species were placed in each vial, with a separate series of vials for each species. Mortality levels were recorded after 3, 7, 14, and 21 days of exposure. After that, the vials were kept for additional 65 days to assess progeny production. Our results indicate that among the tested diatomaceous earth formulations, the application of Silicid resulted in complete control of the major stored-product insect species. Offspring production was noted only for primary colonizers.

**Abstract:**

Diatomaceous earth (DE) formulations are promising alternatives over the use of traditional insecticides. In the present study, a series of laboratory bioassays was carried out to assess the efficacy of three diatomaceous earth formulations, i.e., Silicid, Celatom^®^ MN-23, and SilicoSec^®^, for the control of a wide range of stored-product insect species in soft wheat. The species tested were *Tribolium confusum*, *Tribolium castaneum*, *Sitophilus oryzae*, *Sitophilus granarius*, *Rhyzopertha dominica*, *Oryzaephilus surinamensis*, and *Alphitobious diaperinus*. Different dose rates, i.e., 0 (control), 100, 300, 500, and 1000 ppm, were used for each of the aforementioned dust formulations. Mortality levels of the exposed individuals were assessed after 3, 7, 14, and 21 days of exposure. Moreover, progeny were counted 65 days later. Based on our results, dust formulations were effective for the control of most of the stored-product beetle species tested. Among the DE formulations tested, Silicid could adequately control the stored-product insect species. Complete suppression of offspring was observed only for secondary species (*T. confusum*, *T. castaneum*, *O. surinamensis*, and *A. diaperinus*). For primary species (*S. oryzae*, *S. granarius*, and *R. dominica*), the lowest number of progeny was observed in wheat treated with Silicid. For instance, in the case of *R. dominica*, significantly fewer individuals were produced in Silicid-treated wheat at the highest dose rate. The results of the present study aim to encourage the utilization of DE in stored-product protection as an integrated pest management tool. Additional experimentation is required to apply the tested DE formulations in the field and on different surfaces.

## 1. Introduction

Post-harvest losses due to stored-product insect infestations and deterioration from molds and bacteria correspond to 10% and 20% of the total cereal grain production in developed and developing countries, respectively [1,2]. The annual grain losses during storage in the Indian sub-continent and Sub-Saharan Africa are estimated at 1 and 4 billion dollars, respectively [3]. In some cases, grain losses can reach up to 40% of the harvest due to poor handling and storage methods. For instance, in Tanzania, one of the major maize producers in Sub-Saharan Africa, post-harvest losses of maize account for over 40% of the total production, which is translated to annual losses of around $1.6 billion [2,4]. Therefore, there is an imperative need to control stored-product insect infestations and efficiently protect stored durable commodities. 

Grain protectants have been extensively utilized throughout the globe for the control of stored-product insects [5,6]. Stored-product insect control is mainly based on insecticides such as organophosphates (OPs), pyrethroids, and fumigants, of which the primary insecticide is phosphine. Several studies have reported their insecticidal efficacies against a wide range of target species [7,8,9]. However, the extensive use of the above methods has led to the development of resistance by stored-product insects to insecticides, particularly to phosphine [10,11]. The demand for residue-free food and increased concerns about consumer safety, in conjunction with insect resistance, has prompted researchers to evaluate new substances that are environmentally friendly against stored-product insect species [12,13,14]. 

Inert dusts are promising alternatives to conventional insecticides against numerous stored-product pests [11,15,16,17,18,19,20,21,22]. According to their chemical and physical composition, they are divided into several categories, such as (a) zeolite [22,23,24]; (b) attapulgite [20]; (c) graphene [21]; (d) wood ash [17]; and (e) diatomaceous earth (DE) [25,26,27,28,29,30,31]. One of the well-studied categories is diatomaceous earth, which is registered for direct application on grains in many parts of the world [30]. Diatomaceous earth formulations are based on natural substances originating from the fossils of unicellular algae, i.e., phytoplankton or diatoms [19,32]. The mode of action is not complicated; when the DE is in contact with the insect’s body, it absorbs onto the waxy layer of the insect’s cuticle, resulting in water loss and insect death through desiccation [19,33,34]. DE has low mammalian toxicity, long-term stability, and can be easily removed from treated grains [19]. The main drawback to the use of DE formulations as grain protectants is their negative effect on the physical properties of grain, mainly the bulk density [19,35,36], which is associated with the application of high dose rates of the formulations [16]. 

Numerous studies have revealed the effectiveness of DE for the protection of stored products against a wide range of stored-product insect species [26,29,37,38]. The efficiency of each DE formulation varies remarkably depending on the dose, exposure time, temperature, relative humidity, grain type, and insect life stage (adults or larvae) [28,30,38,39,40]. For instance, Athanassiou et al. [41] investigated different dust formulations at three dose rates against adults of major stored-product insects and they reported high mortality levels even at the lowest dose rate. In addition, the larval stage is considered to be more susceptible to DE than the adult stage [42]. For instance, larvae of the confused flour beetle, *Tribolium confusum* Jacquelin Du Val (Coleoptera: Tenebrionidae), were more susceptible than adults [40]. However, it should be noted that adults of the red flour beetle, *Tribolium castaneum* (Herbst) (Coleoptera: Tenebrionidae), and *T. confusum* are considered to be the least susceptible beetle species to different DE formulations [40,41,42]. Additionally, the particle size of the DE formulation is a very important parameter for achieving the highest efficacy against insects [29,36,43]. For example, smaller particles, i.e., less than 45 μm, were more effective than larger particles against several stored-product beetles [29]. 

Several studies have indicated that diatomaceous earth formulations with different percentages of silicon dioxide and the presence of food additives may give entirely dissimilar mortality rates [26,44]. For instance, Athanassiou et al. [26] investigated the efficacy of two diatomaceous earth formulations, namely Insecto^®^ (containing 86.7% SiO_2_ with 10% food-grade additives) and PyriSec^®^ (containing 88% SiO_2_ with 1.2% natural pyrethrum and 3.1% piperonyl butoxide), after application on wheat and they reported lower mortality levels for Insecto^®^-treated grains than for Pyrisec^®^-treated ones. In this framework, the objective of the present study was to evaluate an available diatomaceous earth formulation (namely Silicid) in comparison with other studied DE formulations against adults of a wide range of stored-product beetle species. Specifically, seven stored-grain beetle species were tested, including three primary colonizers and four secondary colonizers, which are commonly found in stored cereals and other relevant commodities in Greece in various type of facilities [45,46]. 

## 2. Materials and Methods

### 2.1. Insect Species

The species that were tested in our bioassays were *T. confusum*; *T. castaneum*; the rice weevil, *Sitophilus oryzae* (L.) (Coleoptera: Curculionidae); the granary weevil, *Sitophilus granarius* (L.) (Coleoptera: Curculionidae); the lesser grain borer, *Rhyzopertha dominica* (F.) (Coleoptera: Bostrychidae); the saw-toothed grain beetle, *Oryzaephilus surinamensis* (L.) (Coleoptera: Silvanidae); and the lesser mealworm, *Alphitobious diaperinus* (Panzer) (Coleoptera: Tenebrionidae). The primary species were *S. oryzae*, *S. granarius*, and *R. dominica*, and the secondary species were *T. confusum*, *T. castaneum*, *O. surinamensis*, and *A. diaperinus*. The above species were reared at 26 °C under 55% relative humidity and kept in continuous darkness. The rearing substrate for each species was: wheat flour for *T. castaneum* and *T. confusum*; soft wheat for *S. oryzae*, *S. granarius*, and *R. dominica*; oat flakes for *O. surinamensis*; and wheat bran for *A. diaperinus*. Only young adults of mixed sex age were tested in our experiments (<1 month old). The insects used in the experiments were taken from rearings maintained at the Laboratory of Entomology and Agricultural Zoology (LEAZ) at the University of Thessaly for more than 20 years, with the exception of *A. diaperinus*, which has been maintained at LEAZ for around 6 years. 

### 2.2. Diatomaceous Earth Formulations

Three diatomaceous earth formulations (100% natural product) were evaluated in our bioassays against all species: (a) Silicid (BioFa GmbH, Munchigen, Germany), which contains >99% SiO_2_, with <1%; quartz; (b) Celatom^®^ MN-23 (E_P_Minerals^®^, Reno, NV, USA), which contains 83.7% SiO_2_ and Al_2_O_3_ (5.6%), FeO_3_ (2.3%), CaO (0.9%), MgO (0.3%), and other oxides (1.9%); and (c) SilicoSec^®^ (BioFa GmbH, Munchigen, Germany), which contains 90% SiO_2_. All dust formulations were sieved prior to the experiments in order to achieve a uniform particle size (<50 μm). 

### 2.3. Bioassays

The used wheat was uninfected and free of pesticides. Lots of soft wheat (500 g/replicate/dose rate) were dusted with the different DE formulations at four dose rates, i.e., 100, 300, 500, and 1000 ppm (g/kg of grains). One additional series of lots was left untreated and served as the control (0 ppm). The treated wheat was afterwards placed in 1 L glass jars and shaken manually for approximately 2 min to achieve an equal distribution of the dust formulation in the entire wheat mass, with separate jars for each formulation [37]. 

The experimental units for the bioassays were plastic cylindrical vials (3 cm in diameter, 8 cm in height; Carl Roth, Karlsruhe, Germany). The top quarter of the inside of each vial was covered with fluon (polytetrafluoroethylene dispersion; Sigma Aldrich Co., Steinheim am Albuch, Germany) in order to prevent the insects from escaping. After that, twenty grams of soft wheat was filled in each vial. Then, twenty adults of each species were placed in each vial, with a separate series of vials for each species. All vials were kept at the aforementioned conditions (26 °C, 55% relative humidity). Mortality was recorded after 3, 7, 14, and 21 days of exposure. There were three replicates (3 vial replicates for each dust formulation × dose × species combination) and the procedure was repeated three times, preparing new lots of treated and untreated grains each time (3 × 3 = 9 replicates). After the 21-day interval, all adults (dead and alive) were removed individually using a fine paint brush (Lineo, No. 1; Mesko-Pinsel GmbH, Wieseth, Germany). The vials were left at the same conditions for an additional period of 65 days, after which the numbers of progeny were counted. 

### 2.4. Data Analysis

Prior to analysis, all data were tested for normalization and homogeneity using Levene’s test. Where the variance in Levene’s test was not equal, the O’Brien or Brown–Forsythe tests were used. In some cases, data were transformed (exponential, squish, or square root transformation) to follow the assumptions. Control mortality was in all cases very low (<2%), thus no correction was needed. Moreover, control mortality levels were excluded from the statistical analysis. Since the same vials were examined for mortality after 3, 7, 14, and 21 days, mortality data were analyzed separately for each species using repeated multivariate analysis of variance (MANOVA) with dose rate, exposure interval, and DE formulation as the main effects and mortality as the repeated variable. For progeny production, data were submitted to a one-way ANOVA with offspring numbers as the response variable and dust formulation as the main effect. Means were separated using the Tukey-Kramer HSD test at the 5% level. For each species, differences were determined among the dust formulations for each dose rate, including the control. 

## 3. Results

### 3.1. Adult Mortality

For all tested species, all main effects (formulation, dose, time) and associated interactions (formulation × dose, time × formulation, time × dose, time × formulation × dose) were significant since *p* < 0.001 (Table 1). 

### 3.2. Tribolium confusum

The adult mortality levels of *T. confusum* were generally low (Table 2). Significant differences were noted among the tested DE formulations and the four dose rates (100, 300, 500, and 1000 ppm) for each exposure interval (Table 2). Indicatively, after 3 days of exposure, the highest mortality level was recorded for Silicid at the highest dose rate, whereas zero mortality was noted for the rest of the DE formulations tested (Celatom^®^ MN-23 and SilicoSec^®^). After 7 days of exposure, the mortality levels were extremely low (7.8 ± 2.5) even at 500 ppm, whereas the mortality level was 50% at 1000 ppm in the Silicid-treated wheat. After 14 days of exposure, the mortality levels were generally low at 100, 300, and 500 ppm for all formulations tested, with the highest mortality level being found for Silicid. In contrast, at the same exposure interval at the highest dose rate, the mortality level of *T. confusum* adults was almost 100% in the wheat treated with Silicid (Table 2). After 21 days of exposure at 1000 ppm, the mortality levels were 100%, 96%, and 37% for the wheat treated with Silicid, Celatom^®^ MN-23, and SilicoSec^®^, respectively (Table 2).

### 3.3. Tribolium castaneum

In general, low mortality levels were recorded for adults of *T. castaneum* (Table 3). Briefly, the lowest mortality levels (<33%) were recorded for SilicoSec^®^, even at the highest dose rate (1000 ppm) and the longest exposure interval (21 days). In contrast, efficient control was achieved with Silicid (100%) and Celatom^®^ MN-23 (91%) after application of the high dose rate and exposure for 21 days (Table 3). Silicid was very effective against adults of *T. castaneum* even after 7 days of exposure (82% mortality), when the mortality levels for Celatom^®^ MN-23 and SilicoSec^®^ were 38% and 4%, respectively (Table 3). In most cases, significant differences were noted in mortality levels among the dose rates for each exposure interval. When the dose rate was increased, the mortality level was significantly increased in most cases. For instance, after 14 days of exposure, the mortality levels were 6.1%, 7.2%, 21.6%, and 100% at 100, 300, 500, and 1000 ppm in the Silicid-treated wheat, respectively (Table 3). Finally, a noticeable increase in the mortality level of *T. castaneum* adults was recorded with increasing exposure interval (Table 3). 

### 3.4. Sitophilus oryzae

Generally, significant differences were recorded among the three dust formulations for all exposure intervals (Table 4). The mortality levels were extremely low after 3 days of exposure for all formulations tested and no significant differences were noted among the formulations for each dose rate, with the exception of 1000 ppm (Table 4). At longer exposure intervals, the mortality levels continued to rise for all formulations tested and dose rates. For instance, after 7 days of exposure at the highest dose rate (1000 ppm), the mortality level exceeded 80% for Silicid, being significantly different than the respective levels for the other two DE formulations. Significant differences were noted among the tested dose rates after 14 days of exposure, but it must be noted that a mortality level close to 100% was only recorded after application of the highest dose rate of Silicid (Table 4). However, after 21 days of exposure, even at 500 ppm, the mortality levels reached 100%, 97.2%, and 83.3% for Silicid, Celatom^®^ MN-23, and SilicoSec^®^, respectively. 

### 3.5. Sitophilus granarius

Despite the low levels of mortality, there were significant differences among the three formulations tested after 3 days of exposure (Table 5). Specifically, significantly more *S. granarius* adults died when exposed to the wheat treated with 1000 ppm of Silicid, in comparison with the other two formulations tested (Celatom^®^ MN-23 and SilicoSec^®^). The mortality levels were extremely low even after 7 days of exposure, where differences among the dose rates tested were significant only in the case of Silicid (Table 5). However, with the increase in exposure interval, the mortality level was significantly increased. For example, after 21 days of exposure, significant differences were noted among the dose rates and the dust formulations tested. Finally, for individuals of *S. granarius*, complete mortality was noted only after application of 1000 ppm of Silicid after 21 days of exposure (Table 5). 

### 3.6. Rhyzopertha dominica

In general, significant differences were recorded among the dust formulations and the dose rates tested (Table 6). After 3 days, exposure resulted in negligible mortality levels at 100, 300, and 500 ppm. However, at the same exposure interval, a noticeable increase in the adult mortality level was recorded at 1000 ppm of Silicid, which was significantly different in comparison with the other two formulations tested (Table 6). Hence, after 7 days of exposure, almost all exposed *R. dominica* adults were dead (>97% mortality) at the highest dose rate (Table 6). Furthermore, after 14 days of exposure, the mortality levels were 97.2%, 57.2%, and 38.9% in the wheat treated with Silicid, Celatom^®^ MN-23, and SilicoSec^®^, respectively. Finally, it should be noted that complete control was recorded only for Silicid after 21 days of exposure, whereas for Celatom^®^ MN-23 and SilicoSec^®^, the adult mortality levels did not exceed 74% and 55%, respectively (Table 6). 

### 3.7. Oryzaephilus surinamensis

In most cases, significant differences were noted among the three dust formulations and the dose rates tested (Table 7). The mortality level of *O. surinamensis* was highly affected by the exposure interval. Specifically, even after 3 days of exposure, the mortality levels were high, ranging from 73% to 89%, whereas at 1000 ppm, the mortality levels ranged from 93% to 97% (Table 7). Similar results were recorded after 7 and 14 days of exposure. It should be noted that complete control was only noted for SilicoSec^®^, even at 300 ppm after 21 days of exposure (Table 7). 

### 3.8. Alphitobius diaperinus

In most cases, zero mortality was noted for *A. diaperinus* after 3 days of exposure at all doses for Celatom^®^ MN-23 and SilicoSec^®^ (Table 8). Significant differences were noted in the efficacy of the different formulations for the exposure intervals at all dose rates (Table 8). For Silicid, the mortality levels reached 59.4%, 95.0%, 99.5%, and 100% after 3, 7, 14, and 21 days of exposure at the highest dose rate, respectively. Statistically significant differences were noted only for Silicid among the dose rates for each exposure interval, whereas no differences were observed in the mortality levels for Celatom^®^ MN-23 and SilicoSec^®^. In that occasion, the mortality levels were negligible and the individuals of *A. diaperinus* remained unaffected, as the mortality levels did not exceed 17%. Finally, it should be noted that adults of *A. diaperinus* could be easily controlled with Silicid (Table 8). 

### 3.9. Progeny Production

Regarding the offspring production, in most cases, the main effects (except for formulation) as well as all associated interactions were significant, with the exception of *A. diaperinus* (Table 9). Specifically, for *Sitophilus* species, the main effects (formulation, dose) and the associated interaction (formulation × dose) were significant since *p* < 0.005. Moreover, for *Tribolium* species and *O. surinamensis*, only the main effect (dose) played a significant role, whereas for *R. dominica*, the main effect (dose) and the associated interaction (formulation × dose) were significant (Table 9). 

Generally, the numbers of progeny for *T. confusum*, *T. castaneum*, *O. surinamensis*, and *A. diaperinus* after the 65-day incubation period were rather low and did not exceed 1.7 adults per vial (Table 10). In those cases, no significant differences were noted among the three dust formulations tested (Table 10). However, for *S. oryzae*, progeny production was extremely high (up to 278 individuals per vial). Progeny production was noted for all dose rates tested. Specifically, significant differences were noted at 300 and 1000 ppm. More individuals were produced in the wheat treated with the lowest dose rate. At the highest dose rate (1000 ppm), comparing the three dust formulations, significant differences were recorded only between Silicid and SilicoSec^®^ (Table 10). Fewer individuals were produced in the case of *S. granarius*, and the lowest number of progeny was noted for Silicid (20 individuals per vial). Similarly, in the case of *R. dominica*, significantly fewer individuals were produced in the Silicid-treated wheat at the highest dose rate (0.7 individuals per vial). 

## 4. Discussion

The results of the present study clearly demonstrate that there is noticeable variability among the tested stored-product insect beetles regarding their susceptibility to different diatomaceous earth formulations. All diatomaceous earth formulations have the same mode of action [32]. However, the percentage of SiO_2_ as well as the different additives that are used in various DE formulations can result in differences in their efficacy [25,38]. The application of the three DE formulations evaluated in the present study resulted in high mortality levels for most of the stored-product beetles tested, in accordance with the results of previous studies. For instance, SilicoSec^®^ has been previously shown to be effective against adults of *S. oryzae* [25,41,44] and *R. dominica* [47], as well as against larvae of the Mediterranean flour moth, *Ephestia kuehniella* Zeller (Lepidoptera: Pyralidae), and *T. confusum* [38,39,40]. Similarly, our results showed the efficacy of Celatom^®^ MN-23 against adults of *S. oryzae* and *S. granarius*, while Korunic and Fields [48] also reported high efficacy levels against these two species as well as against another curculionid, the maize weevil, *Sitophilus zeamais* Motschulsky (Coleoptera: Curculionidae). Silicid, which had the highest percentage of SiO_2_, was tested for the first time and was the most effective DE formulation against the stored-product insect species tested among the formulations tested. It should be noted that regardless of the formulation, the insect species, dose rate, and exposure time were the main factors (DE origin, form, shape, and size of diatoms) that easily affected the mortality levels of the exposed individuals [49]. 

In general, different stored-product insect species have different susceptibilities to diatomaceous earth [30]. It is generally expected that species with soft and flat bodies are more susceptible to DE, as their cuticle can be easily destroyed through desiccation [30,42,47]. In our study, adults of *O. surinamensis* were very susceptible to the DE formulations already after 7 days of exposure. Similar results were reported by Baliota and Athanassiou [31], who also found high mortality levels for this species at the same exposure interval. Similar findings have also been reported for *C. ferrugineus* [29,32,50]. In contrast, species in the family Tenebrionidae, such as *T. castaneum, T. confusum*, and *A. diaperinus*, are considered tolerant to DE [40,41,42,51]. For instance, Athanassiou et al. [41] found that *T. confusum* adults were less susceptible in comparison with *S. oryzae* and *R. dominica* adults. Much earlier, Carlson and Ball [52] tested the susceptibility of a wide range of stored-product insect species to a commercially available DE formulation and concluded that the most tolerant species were *T. castaneum* and *T. confusum*. In the current study, when Celatom^®^ MN-23 or SilicoSec^®^ was applied in soft wheat, *T. confusum*, *T. castaneum*, and *A. diaperinus* were found to be the least susceptible species. However, when Silicid was applied in soft wheat, most of the tested species were found to be susceptible, since 100% mortality levels were recorded. 

Regarding progeny production, in most cases, offspring was recorded regardless of the DE formulation and dose rate. The high mortality levels that were noted for these species after 21 days of exposure may have resulted from the suppression of progeny production. In general, suppression of progeny production by secondary colonizers is usually expected, as the immature stages develop outside the kernel [53,54,55]. In contrast to the above, internal feeders such as *S. oryzae*, *S. granarius*, and *R. dominica* produced many offspring, mainly at the lowest dose rate. This finding aligned with the results of several previously published papers [15,25]. For instance, Athanassiou et al. [25] noted that progeny production was gradually increased, especially in grain treated with a low DE dose rate. Moreover, the development of the larval stages of internal feeders occurs in the internal part of the kernel; therefore, larvae remain unaffected by agents that are applied on the external part [42,56,57]. In our study, although the DE formulations tested were extremely effective against the parental adults, they were not equally effective in the suppression of progeny production. Nevertheless, in the case of *R. dominica*, Silicid applied in soft wheat at the highest dose rate was more effective in the suppression of progeny production than against the parental individuals. 

A satisfactory level of long-term protection can be easily achieved through the application of high dose rates of diatomaceous earth [32,33]. In the present study, the highest dose of Silicid (1000 ppm), which corresponded to 1 kg of DE formulation/ton of soft wheat, was highly effective against *R. dominica* (98%) and *S. oryzae* (83%) after 7 days of exposure. Similarly, Athanassiou et al. [25] found that 1 kg of SilicoSec^®^/ton of wheat was effective against *S. oryzae.* However, the main drawback to the use of DE formulations is that they are applied at high dose rates, which negatively affects the grain bulk density (volume/weight ratio) [32]. Fields and Korunic [50] mentioned that commercially available DE formulations should be applied at doses between 400 and 1000 ppm. In this context, in some cases, a satisfactory level of protection was provided by the DE formulations tested in the present study after application of 500 ppm. Accordingly, Athanassiou et al. [39] tested a wide range of DE formulations and they found that the mortality levels were >80% for the *S. oryzae* population at the dose rate of 500 ppm. 

## 5. Conclusions

To conclude, our results demonstrate that soft wheat can be satisfactory protected against a wide range of stored-product beetle species with the use of the tested DE formulations. Specifically, the application of Silicid resulted in the complete control of adults of *T. castaneum* and *T. confusum*, the most tolerant species to DE formulations, already after 14 days of exposure. Offspring production was noted only for primary species and not for secondary ones. It should be also mentioned that Silicid provided the highest suppression of progeny production in comparison with the other two DE formulations tested. However, further experimentation is needed in order to evaluate the efficacy of the tested DE formulations under field conditions (i.e., long-term protection of 6 months and impact of bulk density) and in several application scenarios (different surfaces, etc.). 

## Figures and Tables

**Table 1 insects-14-00656-t001:** MANOVA parameters for main effects and associated interactions for the mortality levels of *Tribolium confusum*, *Tribolium castaneum*, *Sitophilus oryzae*, *Sitophilus granarius*, *Rhyzopertha dominica*, *Oryzaephilus surinamensis*, and *Alphitobius diaperinus* adults between and within variables (df_error_ = 96).

Species		*T. confusum*		*T. castaneum*		*S. oryzae*		*S. granarius*		*R. dominica*		*O. surinamensis*		*A. diaperinus*	
Source	df	F	*p*	F	*p*	F	*p*	F	*p*	F	*p*	F	*p*	F	*p*
All between	11	155.51	<0.001	116.54	<0.001	113.71	<0.001	131.87	<0.001	79.60	<0.001	102.34	<0.001	96.61	<0.001
Intercept	1	1090.89	<0.001	927.27	<0.001	3079.38	<0.001	1403.36	<0.001	839.00	<0.001	16,644.30	<0.001	505.81	<0.001
Formulation	2	188.05	<0.001	156.58	<0.001	55.15	<0.001	122.17	<0.001	160.27	<0.001	52.79	<0.001	356.97	<0.001
Dose	3	345.99	<0.001	232.42	<0.001	361.27	<0.001	347.68	<0.001	154.45	<0.001	310.11	<0.001	39.97	<0.001
Time	3	318.25	<0.001	316.35	<0.001	1039.68	<0.001	346.22	<0.001	129.99	<0.001	408.68	<0.001	153.46	<0.001
All within	33	27.40 *	<0.001	23.32 *	<0.001	35.18 *	<0.001	28.26 *	<0.001	8.09 *	<0.001	18.68 *	<0.001	12.69 *	<0.001
Formulation × Dose	6	49.42	<0.001	45.25	<0.001	9.45	<0.001	27.20	<0.001	15.29	<0.001	14.98	<0.001	38.23	<0.001
Time × Formulation	6	36.16 *	<0.001	40.81 *	<0.001	11.30	<0.001	20.59 *	<0.001	8.91 *	<0.001	9.75 *	<0.001	49.67 *	<0.001
Time × Dose	9	60.10 *	<0.001	38.41 *	<0.001	87.19	<0.001	61.07 *	<0.001	8.03 *	<0.001	51.07 *	<0.001	4.59 *	<0.001
Time × Formulation × Dose	18	17.29 *	<0.001	14.52 *	<0.001	16.60 *	<0.001	16.50	<0.001	7.59 *	<0.001	8.68 *	<0.001	4.51 *	<0.001

* Wilks’ lambda approximation was used.

**Table 2 insects-14-00656-t002:** Mortality means (percentage ± SE) of *Tribolium confusum* adults after exposure for 3, 7, 14, and 21 days to soft wheat treated with three diatomaceous earth formulations (Silicid, Celatom^®^ MN-23, and SilicoSec^®^) at four dose rates (100, 300, 500, and 1000 ppm), (df*_total_* = 26).

		Dust Formulations			Statistics	
Exposure Interval	Doses	Silicid	Celatom^®^ MN-23	SilicoSec^®^	F	*p*
3d	100	0.0 ± 0.0 B	0.0 ± 0.0	0.0 ± 0.0	-	-
	300	1.1 ± 0.7 B	0.0 ± 0.0	0.0 ± 0.0	2.28	0.123
	500	2.8 ± 1.2 Ba	0.0 ± 0.0 b	0.0 ± 0.0 b	5.26	0.012
	1000	15.0 ± 1.8 Aa	0.0 ± 0.0 b	0.0 ± 0.0 b	64.80	<0.001
	F	35.230	-	-		
	*p*	<0.001	-	-		
7d	100	0.6 ± 0.6 A	0.6 ± 0.6 B	0.6 ± 0.6	-	-
	300	0.6 ± 0.6 A	0.6 ± 0.6 B	0.6 ± 0.6	-	-
	500	7.8 ± 2.5 Aa	2.7 ± 1.6 Bab	0.6 ± 0.6 b	4.32	0.024
	1000	50.0 ± 4.4 Ba	16.6 ±4.3 Ab	0.6 ± 0.6 c	49.54	<0.001
	F	67.03	10.09	0.33		
	*p*	<0.001	<0.001	0.801		
14d	100	8.3 ± 3.2 Ca	2.7 ± 0.8 Cab	0.6 ± 0.6 Bb	4.18	0.027
	300	11.1 ± 2.6 Ca	1.6 ±0.8 Cb	0.6 ± 0.6 Bb	4.76	0.018
	500	32.2 ± 4.3 Ba	22.2 ±3.5 Ba	0.6 ± 0.6 Bb	24.78	<0.001
	1000	99.4 ± 0.6 Aa	80.6 ± 4.6 Ab	7.2 ± 3.2 Ac	218.13	<0.001
	F	198.54	153.08	4.25		
	*p*	<0.001	<0.001	0.012		
21d	100	11.1 ± 3.5 Ca	3.8 ± 1.1 Cab	0.6 ± 0.6 Bb	6.288	0.006
	300	17.2 ± 3.6 Ca	8.3 ± 2.5 Cab	2.2 ± 1.6 Bb	4.031	0.030
	500	62.7 ± 5.9 Ba	56.1 ± 6.3 Ba	8.3 ± 1.6 Bb	33.74	<0.001
	1000	100.0 ± 0.0 Aa	96.1 ± 2.0 Aa	36.6 ± 6.9 Ab	72.79	<0.001
	F	114.11	147.48	21.00		
	*p*	<0.001	<0.001	<0.001		

For each dust formulation (Silicid, Celatom^®^ MN-23, and SilicoSec^®^) within each column, means followed by the same uppercase letter do not differ significantly according to the Tukey-Kramer HSD test at *p* < 0.05. For each exposure interval and within each dose rate (100, 300, 500, and 1000 ppm), means followed by the same lowercase letter do not differ significantly according to the Tukey-Kramer HSD test at *p* < 0.05. Where no letters exist, no significant differences were noted. For Silicid, at 3d, data were root-transformed and the Brown –Forsythe test was used with F = 2.45 and *p* = 0.08; at 7d, data were squish transformed and the Brown –Forsythe test was used with F = 2.41 and *p* = 0.08; at 14d, no transformation was needed and the O’Brien test was used with F = 2.18 and *p* = 0.10; and at 21d, data were exponentially transformed and the Brown–Forsythe test was used with F = 2.30 and *p* = 0.09. For Celatom^®^ MN-23, at 7d, data were exponentially transformed and the Brown–Forsythe test was used with F = 1.01 and *p* = 0.39; at 14d, no transformation was needed and the O’Brien test was used with F = 2.03 and *p* = 0.12; and at 21d, no transformation was needed and the Brown–Forsythe test was used with F = 2.86 and *p* = 0.05. For SilicoSec^®^, at 7d, no transformation was needed and the Brown–Forsythe test was used with F = 0.33 and *p* = 0.80; at 14d, no transformation was needed and the O’Brien test was used with F = 1.95 and *p* = 0.14; and at 21d, data were exponentially transformed and the Brown–Forsythe test was used with F = 1.0 and *p* = 0.40.

**Table 3 insects-14-00656-t003:** Mortality means (percentage ± SE) of *Tribolium castaneum* adults after exposure for 3, 7, 14, and 21 days to soft wheat treated with three diatomaceous earth formulations (Silicid, Celatom^®^ MN-23, and SilicoSec^®^) at four dose rates (100, 300, 500 and 1000 ppm), (df*_total_* = 26).

		Dust Formulations			Statistics	
Exposure Interval	Doses	Silicid	Celatom^®^ MN-23	SilicoSec^®^	F	*p*
3d	100	0.6 ± 0.6 B	0.0 ± 0.0 B	0.0 ± 0.0	4.00	0.031
	300	2.2 ± 1.6 B	0.6 ± 0.6 B	0.0 ± 0.0	1.27	0.298
	500	8.3 ± 1.6 Ba	0.6 ± 0.6 Bb	0.0 ± 0.0 b	4.00	0.031
	1000	36.6 ± 6.9 Aa	6.6 ± 1.9 Ab	0.6 ± 0.6 b	10.89	<0.001
	F	21.00	6.17	1.00		
	*p*	<0.001	0.002	0.405		
7d	100	3.8 ± 1.8 Ba	0.0 ± 0.0 Bb	0.0 ± 0.0 b	4.55	0.021
	300	5.6 ± 2.2 B	3.8 ± 1.8 B	0.0 ± 0.0	2.87	0.076
	500	8.8 ±2.4 Bab	12.2 ± 3.8 Ba	1.6 ± 0.8 b	4.07	0.030
	1000	81.1 ± 7.2 Aa	37.7 ± 6.4 Ab	3.8 ± 2.3 c	45.05	<0.001
	F	83.95	19.32	2.22		
	*p*	<0.001	<0.001	0.104		
14d	100	6.1 ± 2.3 Ca	5.0 ± 1.1 Cab	0.0 ± 0.0 Bb	4.68	0.019
	300	7.2 ± 1.6 Cb	24.4 ± 5.2 Ba	0.0 ± 0.0 Bb	15.64	<0.001
	500	21.6 ± 3.0 Bb	41.1 ± 4.4 Ba	1.6 ± 0.8 ABc	39.28	<0.001
	1000	100.0 ± 0.0 Aa	61.6 ± 5.5 Ab	5.6 ± 2.5 Ac	179.02	<0.001
	F	463.06	29.07	3.78		
	*p*	<0.001	<0.001	0.019		
21d	100	5.6 ± 2.1 Cb	18.3 ± 2.7 Ca	2.7 ± 1.6 Bb	13.79	<0.001
	300	16.1 ± 4.6 Cb	32.2 ± 5.8 Ca	2.2 ± 1.6 Bb	11.45	<0.001
	500	61.6 ± 6.7 Ba	56.1 ± 4.9 Ba	3.3 ± 1.1 Bb	43.99	<0.001
	1000	100.0 ± 0.0 Aa	91.1 ± 3.5 Aa	32.7 ± 3.9 Ab	144.41	<0.001
	F	106.88	51.43	40.11		
	*p*	<0.001	<0.001	<0.001		

For each dust formulation (Silicid, Celatom^®^ MN-23, and SilicoSec^®^) within each column, means followed by the same uppercase letter do not differ significantly according to the Tukey-Kramer HSD test at *p* < 0.05. For each exposure interval and within each dose rate (100, 300, 500, and 1000 ppm), means followed by the same lowercase letter do not differ significantly according to the Tukey-Kramer HSD test at *p* < 0.05. Where no letters exist, no significant differences were noted. For Silicid, at 3d, data were exponentially transformed and the Brown–Forsythe test was used with F = 1.03 and *p* = 0.392; at 7d, data were root-transformed and the Brown–Forsythe test was used with F = 0.23 and *p* = 0.871; at 14d, data were exponentially transformed and the O’Brien test was used with F = 1.13 and *p* = 0.348; and at 21d, no transformation was needed and the O’Brien test was used with F = 2.41 and *p* = 0.08. For Celatom^®^ MN-23, at 3d, data were exponentially transformed and the Brown–Forsythe test was used with F = 1.04 and *p* = 0.385; at 7d, data were exponentially transformed and the Brown–Forsythe test was used with F = 1.01 and *p* = 0.389; at 14d, no transformation was needed and the O’Brien test was used with F = 1.91 and *p* = 0.14; and at 21d, no transformation was needed and the Brown–Forsythe test was used with F = 2.56 and *p* = 0.07. For SilicoSec^®^, no transformation was needed, and at 3d, the Brown–Forsythe test was used with F = 1.00 and *p* = 0.40; at 7d, the O’Brien test was used with F = 2.03 and *p* = 0.12; at 14d, the O’Brien test was used with F = 1.34 and *p* = 0.278; and at 21d, the Brown–Forsythe test was used with F = 2.35 and *p* = 0.09.

**Table 4 insects-14-00656-t004:** Mortality means (percentage ± SE) of *Sitophilus oryzae* adults after exposure for 3, 7, 14, and 21 days to soft wheat treated with three diatomaceous earth formulations (Silicid, Celatom^®^ MN-23, and SilicoSec^®^) at four dose rates (100, 300, 500, and 1000 ppm), (df*_total_* = 26).

		Dust Formulations			Statistics	
Exposure Interval	Doses	Silicid	Celatom^®^ MN-23	SilicoSec^®^	F	*p*
3d	100	2.2 ± 0.8 B	1.1 ± 0.7 B	2.2 ± 1.2 B	0.44	0.646
	300	1.6 ± 0.8 Bb	6.6 ± 2.2 Bab	2.2 ± 1.2 Ba	4.00	0.031
	500	6.1 ± 1.1 Bab	8.3 ± 2.5 Ba	2.2 ± 1.2 Bb	3.20	0.058
	1000	33.9 ± 6.2 Aa	20.6 ± 2.6 Aa	5.0 ± 1.4 Ab	12.86	<0.001
	F	22.53	14.21	3.61		
	*p*	<0.001	<0.001	0.023		
7d	100	2.7 ± 1.2 C	2.7 ± 1.2 C	6.1 ± 1.6 B	2.00	0.157
	300	3.9 ± 1.1 C	10.0 ± 3.2 B	8.9 ± 1.6 AB	2.22	0.129
	500	31.1 ± 5.3 Ba	20.6 ± 3.0 Ba	5.0 ± 1.1 Bb	13.02	<0.001
	1000	82.2 ± 8.3 Aa	55.6 ± 5.9 Ab	12.7 ± 1.8 Ac	33.62	<0.001
	F	54.30	38.43	4.71		
	*p*	<0.001	<0.001	0.007		
14d	100	6.6 ± 1.6 C	8.3 ± 2.3 D	7.7 ± 2.2 C	0.16	0.850
	300	29.5 ± 4.8 B	33.3 ± 3.7 C	23.9 ± 3.9 BC	1.27	0.297
	500	96.1 ± 1.1 Aa	71.6 ± 4.5 Bb	33.3 ± 6.6 Bc	45.16	<0.001
	1000	98.9 ± 1.1 Aa	96.6 ± 1.5 Aa	81.1 ± 3.7 Ab	16.50	<0.001
	F	300.55	145.92	50.85		
	*p*	<0.001	<0.001	<0.001		
21d	100	16.6 ± 3.5 B	15.0 ± 3.0 C	10.6 ± 2.8 B	1.01	0.377
	300	95.0 ± 2.7 Aa	65.6 ± 5.0 Bb	49.4 ± 6.7 Bb	20.24	<0.001
	500	100.0 ± 0.0 Aa	97.2 ± 1.6 Aa	83.3 ± 5.0 Ab	8.37	<0.001
	1000	100.0 ± 0.0 A	99.4 ± 0.5 A	100.0 ± 0.0 A	1.00	0.382
	F	332.27	165.36	78.50		
	*p*	<0.001	<0.001	<0.001		

For each dust formulation (Silicid, Celatom^®^ MN-23, and SilicoSec^®^) within each column, means followed by the same uppercase letter do not differ significantly according to the Tukey-Kramer HSD test at *p* < 0.05. For each exposure interval and within each dose rate (100, 300, 500, and 1000 ppm), means followed by the same lowercase letter do not differ significantly according to the Tukey-Kramer HSD test at *p* < 0.05. Where no letters exist, no significant differences were noted. For Silicid, at 3d, data were exponentially transformed and the Brown–Forsythe test was used with F = 1.03 and *p* = 0.392; at 7d, no transformation was needed and the O’ Brien test was used with F = 1.79 and *p* = 0.167; at 14d, data were exponentially transformed and the Brown–Forsythe test was used with F = 2.46 and *p* = 0.080; and at 21d, no transformation was needed and the O’Brien test was used with F = 2.36 and *p* = 0.089. For Celatom^®^ MN-23, at 3d, no transformation was needed and the O’ Brien test was used with F = 1.57 and *p* = 0.215; at 7d, data were exponentially transformed and the Brown–Forsythe test was used with F = 2.30 and *p* = 0.099; at 14d, no transformation was needed and the Brown–Forsythe test was used with F = 2.42 and *p* = 0.08; and at 21d, no transformation was needed and the O’Brien test was used with F = 2.07 and *p* = 0.12. For SilicoSec^®^, at 3d, no transformation was needed and Levene’s test was used with F = 0.07 and *p* = 0.97; at 7d, no transformation was needed and Levene’s test was used with F = 1.14 and *p* = 0.346; at 14d, no transformation was needed and the Brown–Forsythe was used with F = 2.06 and *p* = 0.064; and at 21d, data were exponentially transformed and the Brown–Forsythe test was used with F = 2.0 and *p* = 0.095.

**Table 5 insects-14-00656-t005:** Mortality means (percentage ± SE) of *Sitophilus granarius* adults after exposure for 3, 7, 14, and 21 days to soft wheat treated with three diatomaceous earth formulations (Silicid, Celatom^®^ MN-23, and SilicoSec^®^) at four dose rates (100, 300, 500, and 1000 ppm), (df*_total_* = 26).

		Dust Formulations			Statistics	
Exposure Interval	Doses	Silicid	Celatom^®^ MN-23	SilicoSec^®^	F	*p*
3d	100	2.7 ± 1.2 C	2.2 ± 0.8	1.1 ± 0.7	0.77	0.470
	300	10.0 ± 1.6 BCa	1.6 ± 0.8 b	2.7 ± 1.2 b	12.43	<0.001
	500	12.2 ± 2.3 Ba	2.7 ± 1.2 b	1.6 ± 1.1 b	11.89	<0.001
	1000	23.8 ± 3.6 Aa	5.0 ± 1.4 b	0.6 ± 0.6 b	29.84	<0.001
	F	13.39	1.70	0.97		
	*p*	<0.001	0.186	0.417		
7d	100	3.9 ± 1.6 B	2.7 ± 0.8	1.6 ± 0.8	0.90	0.417
	300	9.4 ± 1.7 Ba	2.7 ± 1.2 b	3.3 ± 0.8 b	7.82	0.002
	500	12.7 ± 2.6 Ba	3.9 ± 2.0 b	2.2 ± 1.2 b	7.72	0.002
	1000	42.7 ± 5.5 Aa	6.1 ± 1.8 b	2.2 ±0.8 b	43.26	<0.001
	F	28.01	1.03	0.53		
	*p*	<0.001	0.391	0.659		
14d	100	4.4 ± 1.5 C	3.3 ± 1.1 B	2.7 ± 0.8 C	0.47	0.627
	300	10.5 ± 2.1 Ca	3.9 ± 1.3 Bb	6.1 ± 1.6 Cab	3.82	0.036
	500	26.1 ± 4.4 Ba	6.1 ± 1.8 Bb	22.2 ± 4.4 Ba	7.87	0.002
	1000	93.9 ± 1.6 Aa	14.4 ± 2.4 Ac	64.4 ± 5.6 Ab	120.84	<0.001
	F	229.42	8.46	58.96		
	*p*	<0.001	<0.001	<0.001		
21d	100	5.0 ± 1.4 D	2.7 ± 0.8 B	2.7 ± 1.2 C	1.14	0.335
	300	16.6 ± 3.2 Ca	5.0 ± 1.4 Bb	10.5 ± 2.8 Cab	4.99	0.015
	500	78.3 ± 4.7 Ba	13.9 ± 3.3 Bb	63.3 ± 4.6 Ba	62.35	<0.001
	1000	100.0 ± 0.0 Aa	73.3 ± 8.2 Ab	99.4 ± 0.6 Aa	10.18	<0.001
	F	247.25	54.51	267.33		
	*p*	<0.001	<0.001	<0.001		

For each dust formulation (Silicid, Celatom^®^ MN-23, and SilicoSec^®^) within each column, means followed by the same uppercase letter do not differ significantly according to the Tukey-Kramer HSD test at *p* < 0.05. For each exposure interval and within each dose rate (100, 300, 500, and 1000 ppm), means followed by the same lowercase letter do not differ significantly according to the Tukey-Kramer HSD test at *p* < 0.05. Where no letters exist, no significant differences were noted. For Silicid, at 3d, no transformation was needed and Levene’s test was used with F = 2.15 and *p* = 0.112; at 7d, data were exponentially transformed and the Brown–Forsythe test was used with F = 1.0 and *p* = 0.405; at 14d, data were exponentially transformed and the Brown–Forsythe test was used with F = 2.345 and *p* = 0.09; and at 21d, data were exponentially transformed and the Brown–Forsythe test was used with F = 2.32 and *p* = 0.093. For Celatom^®^ MN-23, at 3d, no transformation was needed and Levene’s test was used with F = 1.01 and *p* = 0.396; at 7d, no transformation was needed and the Brown–Forsythe test was used with F = 0.56 and *p* = 0.640; at 14d, data were exponentially transformed and the Brown–Forsythe test was used with F = 2.34 and *p* = 0.09; and at 21d, data were exponentially transformed and the Brown–Forsythe test was used with F = 1.0 and *p* = 0.40. For SilicoSec^®^, at 3d, no transformation was needed and the O’Brien test was used with F = 0.86 and *p* = 0.46; at 7d, no transformation was needed and Levene’s test was used with F = 0.91 and *p* = 0.44; at 14d, data were exponentially transformed and the Brown–Forsythe test was used with F = 2.28 and *p* = 0.09; and at 21d, data were exponentially transformed and the Brown–Forsythe test was used with F = 1.0 and *p* = 0.405.

**Table 6 insects-14-00656-t006:** Mortality means (percentage ± SE) of *Rhyzopertha dominica* adults after exposure for 3, 7, 14, and 21 days to soft wheat treated with three diatomaceous earth formulations (Silicid, Celatom^®^ MN-23, and SilicoSec^®^) at four dose rates (100, 300, 500, and 1000 ppm), (df*_total_* = 26).

		Dust Formulations			Statistics	
Exposure Interval	Doses	Silicid	Celatom^®^ MN-23	SilicoSec^®^	F	*p*
3d	100	7.2 ± 2.2 Ca	0.6 ± 0.6 Bb	0.6 ± 0.6 b	8.0	<0.001
	300	19.4 ± 3.5 Ba	1.6 ± 0.8 Bb	0.6 ± 0.6 b	24.40	<0.001
	500	21.1 ± 3.5 Ba	1.1 ± 1.1 Bb	1.1 ± 0.7 b	28.32	<0.001
	1000	90.0 ± 2.0 Aa	18.3 ± 4.5 Ab	5.0 ± 2.3 c	205.18	<0.001
	F	164.63	12.89	2.74		
	*p*	<0.001	<0.001	0.059		
7d	100	17.7 ± 4.0 Ca	3.3 ± 1.6 Bb	0.6 ± 0.6 Bb	13.40	<0.001
	300	39.4 ± 6.7 Ba	5.0 ± 1.1 Bb	2.7 ± 1.2 Bb	25.87	<0.001
	500	35.5 ± 6.1 BCa	3.3 ± 2.2 Bb	1.1 ± 0.7 Bb	25.80	<0.001
	1000	97.2 ± 1.8 Aa	43.3 ± 4.7 Ab	18.3 ± 4.4 Ac	104.62	<0.001
	F	45.88	48.78	12.80		
	*p*	<0.001	<0.001	<0.001		
14d	100	17.7 ± 4.0 Ca	11.6 ± 4.0 Bab	0.5 ± 0.6 Bb	6.92	0.004
	300	39.4 ± 6.7 Ba	10.5 ± 2.6 Bb	6.1 ± 2.1 Bb	16.90	<0.001
	500	35.5 ± 6.1 BCa	10.0 ± 3.9 Bb	2.7 ± 1.2 Bb	16.10	<0.001
	1000	97.2 ± 1.8 A a	57.2 ± 4.3 Ab	38.9 ± 6.6 Ac	40.04	<0.001
	F	45.88	36.82	25.57		
	*p*	<0.001	<0.001	<0.001		
21d	100	20.0 ± 5.6 Da	24.4 ± 4.7 Ba	3.8 ± 1.3 Bb	6.22	<0.001
	300	48.3 ± 6.1 Ca	21.6 ± 3.5 Bb	16.1 ± 3.0 Bb	14.7	<0.001
	500	70.5 ± 4.4 Ba	14.4 ±2.5 Bb	15.0 ± 3.6 Bb	78.91	<0.001
	1000	100.0 ± 0.0 Aa	73.3 ± 4.4 Ab	54.4 ±7.5 Ac	20.30	<0.001
	F	51.13	46.91	23.81		
	*p*	<0.001	<0.001	<0.001		

For each dust formulation (Silicid, Celatom^®^ MN-23, and SilicoSec^®^) within each column, means followed by the same uppercase letter do not differ significantly according to the Tukey-Kramer HSD test at *p* < 0.05. For each exposure interval and within each dose rate (100, 300, 500, and 1000 ppm), means followed by the same lowercase letter do not differ significantly according to the Tukey-Kramer HSD test at *p* < 0.05. Where no letters exist, no significant differences were noted. For Silicid, no transformation was needed, and at 3d, Levene’s test was used with F = 1.53 and *p* = 0.224; at 7d, Levene’s test was used with F = 1.84 and *p* = 0.159; at 14d, Levene’s test was used with F = 1.84 and *p* = 0.159; and at 21d, the O’Brien test was used with F = 1.717 and *p* = 0.172. For Celatom^®^ MN-23, at 3d, data were exponentially transformed and the Brown–Forsythe test was used with F = 1.01 and *p* = 0.405; at 7d, no transformation was needed and the O’Brien test was used with F = 2.650 and *p* = 0.065; at 14d, no transformation was needed and Levene’s test was used with F = 0.715 and *p* = 0.550; and at 21d, no transformation was needed and the O’Brien test was used with F = 2.29 and *p* = 0.096. For SilicoSec^®^, at 3d, no transformation was needed and the Brown–Forsythe test was used with F = 2.74 and *p* = 0.059; at 7d, 14d, and 21d, data were exponentially transformed and the Brown–Forsythe test was used with 7d: F = 1.01, *p* = 0.398, 14d: F = 1.0, *p* = 0.405; and 21d: F = 1.03, *p* = 0.392.

**Table 7 insects-14-00656-t007:** Mortality means (percentage ± SE) of *Oryzaephilus surinamensis* adults after exposure for 3, 7, 14, and 21 days to soft wheat treated with three diatomaceous earth formulations (Silicid, Celatom^®^ MN-23, and SilicoSec^®^) at four dose rates (100, 300, 500, and 1000 ppm), (df*_total_* = 26).

		Dust Formulations			Statistics	
Exposure Interval	Doses	Silicid	Celatom^®^ MN-23	SilicoSec^®^	F	*p*
3d	100	21.6 ± 4.4 Ca	2.2 ± 1.2 Cb	12.7 ± 2.6 Cab	10.17	<0.001
	300	56.1 ± 4.8 Ba	7.7 ± 2.9 Cb	47.2 ± 8.1 Ba	20.13	<0.001
	500	88.9 ± 2.6 Aa	73.3 ± 4.1 Bb	79.4 ± 4.1 Aab	4.48	0.022
	1000	93.3 ± 2.2 A	96.6 ± 1.4 A	95.6 ± 2.4 A	0.67	0.518
	F	81.39	304.11	55.61		
	*p*	<0.001	<0.001	<0.001		
7d	100	46.6 ± 4.1 Ca	10.0 ± 2.2 Cb	35.6 ± 6.0 Ba	18.08	<0.001
	300	82.7 ± 1.8 Ba	38.3 ± 7.1 Bb	80.6 ± 6.2 Ab	20.28	<0.001
	500	98.3 ± 0.8 A	93.9 ± 2.0 A	93.9 ± 2.8 A	1.53	0.236
	1000	98.5 ± 4.1 A	96.7 ± 1.4 A	97.2 ± 1.8 A	1.26	0.300
	F	82.23	118.12	37.30		
	*p*	<0.001	<0.001	<0.001		
14d	100	86.6 ± 2.3 Ca	42.8 ± 4.8 Bb	83.9 ± 4.6 Ba	35.31	<0.001
	300	92.2 ±1.4 BCab	89.5 ± 2.2 Ab	97.7 ± 1.4 Aa	5.69	<0.001
	500	99.4 ± 0.6 A	98.9 ± 1.1 A	97.2 ± 1.4 A	0.11	0.895
	1000	96.6 ± 1.1 AB	97.8 ± 1.4 A	97.8 ± 1.8 A	0.130	0.878
	F	13.23	87.96	6.89		
	*p*	<0.001	<0.001	<0.001		
21d	100	89.4 ± 2.2 Cb	74.5 ± 3.2 Bc	97.7 ± 0.8 Ba	25.16	<0.001
	300	92.7 ± 1.2 BCb	96.1 ± 2.0 Aab	100.0 ± 0.0 Aa	7.15	0.003
	500	99.4 ± 0.6 A	98.9 ± 1.1 A	100.0 ± 0.0 A	0.600	0.556
	1000	97.8 ± 1.2 AB	97.8 ± 1.4 A	100.0 ± 0.0 A	1.36	0.275
	F	10.01	29.83	6.40		
	*p*	<0.001	<0.001	<0.001		

For each dust formulation (Silicid, Celatom^®^ MN-23, and SilicoSec^®^) within each column, means followed by the same uppercase letter do not differ significantly according to the Tukey-Kramer HSD test at *p* < 0.05. For each exposure interval and within each dose rate (100, 300, 500, and 1000 ppm), means followed by the same lowercase letter do not differ significantly according to the Tukey-Kramer HSD test at *p* < 0.05. Where no letters exist, no significant differences were noted. For Silicid, at 3d, no transformation was needed and the O’Brien test was used with F = 1.69 and *p* = 0.188; at 7d, no transformation was needed and the Brown–Forsythe test was used with F = 2.65 and *p* = 0.065; at 14d, data were exponentially transformed and the O’Brien test was used with F = 2.35 and *p* = 0.090; and at 21d, no transformation was needed and the Brown–Forsythe test was used with F = 2.53 and *p* = 0.074. For Celatom^®^ MN-23, at 3d, no transformation was needed and the Brown–Forsythe test was used with F = 2.28 and *p* = 0.090; at 7d, no transformation was needed and the O’Brien test was used with F = 2.40 and *p* = 0.086; at 14d, data were exponentially transformed and the Brown–Forsythe test was used with F = 0.814 and *p* = 0.495; and at 21d, no transformation was needed and the O’Brien test was used with F = 2.23 and *p* = 0.103. For SilicoSec^®^, at 3d, 7d, 14d, and 21d, data were exponentially transformed and the Brown–Forsythe test was used for 3d: F = 2.46 and *p* = 0.080, 7d: F = 1.78 and *p* = 0.170, 14d: F = 0.160 and F = 0.921, and 21d: F = 1.03 and *p* = 0.392.

**Table 8 insects-14-00656-t008:** Mortality means (percentage ± SE) of *Alphitobius diaperinus* adults after exposure for 3, 7, 14, and 21 days to soft wheat treated with three diatomaceous earth formulations (Silicid, Celatom^®^ MN-23, and SilicoSec^®^) at four dose rates (100, 300, 500, and 1000 ppm), (df*_total_* = 26).

		Dust Formulations			Statistics	
Exposure Interval	Doses	Silicid	Celatom^®^ MN-23	SilicoSec^®^	F	*p*
3d	100	6.6 ± 1.6 Ba	0.0 ± 0.0 b	0.0 ± 0.0 b	13.30	<0.001
	300	10.0 ± 3.4 Ba	0.0 ± 0.0 b	0.0 ± 0.0 b	8.47	<0.001
	500	10.0 ± 2.5 Ba	0.0 ± 0.0 b	0.0 ± 0.0 b	16.00	<0.001
	1000	59.4 ± 6.6 Aa	0.6 ± 0.6 b	0.0 ± 0.0 b	77.69	<0.001
	F	39.12	1.00	1.00		
	*p*	<0.001	0.405	0.405		
7d	100	20.6 ± 4.2 Ba	0.6 ± 0.6 b	0.6 ± 0.6 Bb	21.07	<0.001
	300	23.9 ± 5.9 Ba	0.0 ± 0.0 b	0.0 ± 0.0 Bb	16.18	<0.001
	500	25.0 ± 7.0 Ba	0.0 ± 0.0 b	0.0 ± 0.0 Bb	12.67	0.002
	1000	95.0 ± 3.3 Aa	1.1 ± 0.7 b	3.3 ±1.1 Ab	660.37	<0.001
	F	45.39	1.33	6.00		
	*p*	<0.001	0.280	0.002		
14d	100	31.1 ± 4.9 Ba	2.7 ± 1.2 b	4.4 ± 1.5 b	27.07	<0.001
	300	40.6 ± 8.0 Ba	2.2 ± 1.2 b	2.7 ± 1.2 b	21.13	<0.001
	500	40.6 ± 6.9 Ba	0.6 ± 0.6 b	0.6 ± 0.6 b	32.75	<0.001
	1000	99.5 ± 0.6 Aa	1.1 ±0.7 b	1.1 ± 0.7 b	6962.00	<0.001
	F	28.40	1.08	2.62		
	*p*	<0.001	0.368	0.067		
21d	100	47.7 ± 4.7 Ba	2.7 ± 1.2 c	16.1 ± 2.3 b	53.67	<0.001
	300	57.7 ± 7.4 Ba	2.2 ± 0.7 b	16.6 ± 2.8 b	38.50	<0.001
	500	68.9 ± 6.8 Ba	0.6 ± 0.6 b	10.0 ± 1.6 b	82.01	<0.001
	1000	100.0 ±0.0 Aa	1.1 ±0.7 c	17.7 ± 2.6 b	1112.00	<0.001
	F	16.39	1.08	2.07		
	*p*	<0.001	0.368	0.122		

For each dust formulation (Silicid, Celatom^®^ MN-23, and SilicoSec^®^) within each column, means followed by the same uppercase letter do not differ significantly according to the Tukey-Kramer HSD test at *p* < 0.05. For each exposure interval and within each dose rate (100, 300, 500, and 1000 ppm), means followed by the same lowercase letter do not differ significantly according to the Tukey-Kramer HSD test at *p* < 0.05. Where no letters exist, no significant differences were noted. For Silicid, no transformation was needed, and at 3d, the Brown–Forsythe test was used with F = 2.45 and *p* = 0.081; at 7d, Levene’s test was used with F = 1.02 and *p* = 0.395; at 14d, the O’Brien test was used with F = 1.54 and *p* = 0.221; and at 21d, the O’Brien test was used with F = 2.26 and *p* = 0.100. For Celatom^®^ MN-23, no transformation was needed, and at 3d, the Brown–Forsythe test was used with F = 1.00 and *p* = 0.405; at 7d, the Brown–Forsythe test was used with F = 1.33 and *p* = 0.280; at 14d, the Brown–Forsythe test was used with F = 1.08 and *p* = 0.368; and at 21d, the Brown–Forsythe test was used with F = 1.08 and *p* = 0.368. For SilicoSec^®^, at 3d, no transformation was needed and the O’Brien test was used with F = 1.13 and *p* = 0.348; at 7d, data were exponentially transformed and the Brown–Forsythe test was used with F = 1.05 and *p* = 0.381; at 14d, data were exponentially transformed and the Brown–Forsythe test was used with F = 2.49 and *p* = 0.077; and at 21d, no transformation was needed and Levene’s test was used with F = 1.24 and *p* = 0.308.

**Table 9 insects-14-00656-t009:** MANOVA parameters for main effects and associated interactions for progeny production of *Tribolium confusum*, *Tribolium castaneum*, *Sitophilus oryzae*, *Sitophilus granarius*, *Rhyzopertha dominica*, *Oryzaephilus surinamensis*, and *Alphitobius diaperinus* adults between and within variables (df_error_ = 120).

Species		*T. confusum*		*T. castaneum*		*S. oryzae*		*S. granarius*		*R. dominica*		*O. surinamensis*		*A. diaperinus*	
Source	df	F	*p*	F	*p*	F	*p*	F	*p*	F	*p*	F	*p*	F	*p*
Whole model	14	33.65	<0.001	3.54	<0.001	4.40	<0.001	6.35	<0.001	2.97	<0.001	6.18	<0.001	0.0	1.00
Intercept	1	143.81	<0.001	34.43	<0.001	866.22	<0.001	764.65	<0.001	156.57	<0.001	31.86	<0.001	0.0	1.00
Formulation	2	0.11	0.891	0.40	0.664	5.76	0.004	5.36	0.005	0.517	0.597	0.13	0.869	0.0	1.00
Dose	4	117.60	<0.001	10.90	<0.001	5.01	<0.001	12.45	<0.001	4.19	0.003	21.36	<0.001	0.0	1.00
Formulation × Dose	8	0.065	0.999	0.64	0.736	3.75	<0.001	3.54	0.001	2.98	0.004	0.11	0.998	0.0	1.00

**Table 10 insects-14-00656-t010:** Μeans (adults per vial ± SE) of progeny production of *Tribolium confusum*, *Tribolium castaneum*, *Sitophilus oryzae*, *Sitophilus granarius*, *Rhyzopertha dominica*, *Oryzaephilus surinamensis*, and *Alphitobius diaperinus* adults after 65 days in soft wheat treated with different diatomaceous earth formulations at five doses (0, 100, 300, 500, and 1000 ppm), (df_total_ = 26).

		Doses				
Species	Formulations	0 ppm (Control)	100 ppm	300 ppm	500 ppm	1000 ppm
*Tribolium confusum*	Silicid	22.6 ± 3.4	0.0 ± 0.0	0.0 ± 0.0	0.6 ± 0.3	0.0 ± 0.0
	Celatom^®^ MN-23	22.6 ± 3.4	1.4 ± 0.6	0.0 ± 0.0	0.6 ± 0.3	0.0 ± 0.0
	SilicoSec^®^	22.6 ± 3.4	1.2 ± 0.5	0.0 ± 0.0	0.6 ± 0.1	0.0 ± 0.0
*Tribolium castaneum*	Silicid	2.2 ± 2.2	0.6 ± 0.4	0.2 ± 0.2	0.3 ± 0.3	0.0 ± 0.0
	Celatom^®^ MN-23	2.2 ± 2.2	1.7 ± 0.9	0.0 ± 0.0	0.3 ±0.1	0.0 ± 0.0
	SilicoSec^®^	2.2 ± 2.2	0.2 ± 0.2	0.3 ± 0.1	0.2 ± 0.1	0.0 ± 0.0
*Sitophilus oryzae*	Silicid	164.0 ± 24.9	232.2 ± 46.3	231.9 ± 27.2 a	201.0 ± 13.8	98.6 ± 21.0 b
	Celatom^®^ MN-23	164.0 ± 24.9	212.7 ± 20.4	82.0 ± 17.8 b	238.9 ± 29.6	199.2 ± 21.5 a
	SilicoSec^®^	164.0 ± 24.9	277.4 ± 29.9	230.3 ± 27.7 a	244.6 ± 28.7	237.9 ± 16.8 a
*Sitophilus granarius*	Silicid	140.4 ± 13.5	89.2 ± 17.6 b	107.1 ± 13.2	72.1 ± 23.2	20.7 ± 2.7 c
	Celatom^®^ MN-23	140.4 ± 13.5	113.9 ± 8.7 ab	110.6 ± 7.6	83.6 ± 14.8	126.9 ± 22.9 a
	SilicoSec^®^	140.4 ± 13.5	140.4 ± 8.4 a	81.2 ± 17.4	82.7 ± 8.7	56.6 ± 6.7 b
*Rhyzopertha dominica*	Silicid	66.2 ± 15.0	32.2 ± 6.1	75.0 ±26.3 a	46.7 ± 11.4	0.7 ± 0.6 b
	Celatom^®^ MN-23	66.2 ± 15.0	49.0 ± 19.8	14.1 ± 2.7 b	34.6 ± 12.9	53.7 ± 10.3 a
	SilicoSec^®^	66.2 ± 15.0	49.6 ± 12.9	19.3 ± 6.1 b	25.5 ± 5.6	23.1 ± 3.6 b
*Oryzaephilus surinamensis*	Silicid	7.4 ± 2.0	0.9 ± 0.3	0.0 ± 0.0	0.0 ± 0.0	0.0 ± 0.0
	Celatom^®^ MN-23	7.4 ± 2.0	1.1 ± 0.4	0.0 ± 0.0	0.0 ± 0.0	0.0 ± 0.0
	SilicoSec^®^	7.4 ± 2.0	0.3 ± 1.6	0.0 ± 0.0	0.0 ± 0.0	0.0 ± 0.0
*Alphitobius diaperinus*	Silicid	0.0 ± 0.0	0.0 ± 0.0	0.0 ± 0.0	0.0 ± 0.0	0.0 ± 0.0
	Celatom^®^ MN-23	0.0 ± 0.0	0.0 ± 0.0	0.0 ± 0.0	0.0 ± 0.0	0.0 ± 0.0
	SilicoSec^®^	0.0 ± 0.0	0.0 ± 0.0	0.0 ± 0.0	0.0 ± 0.0	0.0 ± 0.0

For each dose rate and species, ANOVA parameters: at 100 ppm, for *S. granarius* F = 4.26, *p* = 0.026; at 300 ppm, for *S. oryzae* F = 12.13, *p* < 0.001, and for *R. dominica* F = 4.60, *p* = 0.020; at 1000 ppm, for *S. oryzae* F = 13.06, *p* < 0.001, for *S. granarius* F = 15.06, *p* < 0.001, and for *R. dominica* F = 17.46, *p* < 0.001. Where no letters exist, no significant differences were recorded. One set of vials was used for the control for all dust formulations.

## Data Availability

The data presented in this study are available on request from the corresponding author.

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
