# Peer review of "Insecticidal Effect of Diatomaceous Earth Formulations for the Control of a Wide Range of Stored-Product Beetle Species"

_insects, 2023, doi:10.3390/insects14070656_

Round 1

Reviewer 1 Report

Authors presented a study of the evaluation of diatomaceous earth formulation Silicid against stored product insect species, and gave a comparison of its activity with two commercial DE formulations. Overall, the manuscript is clear and presented in well-structured manner. The introduction is relevant and theory based. The methods are appropriate, although clarification of some details should be provided (specific comments are listed within the attached file). Overall, the results are clear and good interpreted. The discussion needs to be improved (see attached file). The merit of this study is in the basic information about the insecticidal activity of Silicid against stored product insects which offer new opportunities for further investigations of DE (Silicid) usage in practice. The authors should give some detailed information of the future testing with Silicid (long term protection, impact on bulk density ...). 

Author Response

Reviewer 1

Authors presented a study of the evaluation of diatomaceous earth formulation Silicid against stored product insect species, and gave a comparison of its activity with two commercial DE formulations. Overall, the manuscript is clear and presented in well-structured manner. The introduction is relevant and theory based. The methods are appropriate, although clarification of some details should be provided (specific comments are listed within the attached file). Overall, the results are clear and good interpreted. The discussion needs to be improved (see attached file). The merit of this study is in the basic information about the insecticidal activity of Silicid against stored product insects which offer new opportunities for further investigations of DE (Silicid) usage in practice. The authors should give some detailed information of the future testing with Silicid (long term protection, impact on bulk density ...).

REPLY: The authors would like to thank the reviewer for the comments and recommendations on improving the quality of the manuscript before resubmission. A revised version is submitted with the proposed corrections/additions addressed. A point by point reply on each comment/correction can be found below. We have included this information in the last paragraph (discussion section), please see lines 383- 384.

Abstract: it should be indicated on which type of grain the DE formulations were tested. It is already explained in the section Material and method, but it should also be visible in the Abstract.

REPLY: Correct, we have included the type of grain in the Abstract. Please see line 26.

Line 14 use the name SilicoSec® instead of SilicoSec and make changes through the whole manuscript where it is mentioned

REPLY: Thank you for this comment, we have revised it throughout the text.

Line 25 Add: wheat treated with, after observed in

REPLY: Revised, please see line 38.

Line 50/51 Reorder the sentence as: …has turn the researchers to the evaluation of new substances against stored-product insects species, which are environmental friendly

REPLY: Done, please see line 64.

Line 60 DE absorbs the waxy layer of the insect cuticule, not vice versa. Please change the statement.

REPLY: Revised. Please see line 75-76.

Lie 74 and 75 Please check the letter reorder in the name Coleoptera. And also check it through the whole manuscript, since I spotted in several other places

REPLY: Correct. We have changed “Coleoptrera” to “Coleoptera” throughout the text.

Line 131 Are those conditions under which you kept the vials, the same as conditions for incest rearing or some other? Please indicate that, because it is not clear from the way it is written

REPLY: We have added the conditions in the parenthesis “(26oC, 55% relative humidity). Please see lines 152-153.

Line 329 Add after 21 days of exposure…..after 300 ppm

REPLY: Revised, please see line 252.

Line 437 Besides that, some other aspects are relevant for DEs activity. Like DE origin, forms, shapes and size of diatoms. For example, diatom species with flat cell walls better cover insect cuticula then diatom species with round or cylindrical shape, so their ability to absorb insects cuticular wax could be higher which increases water loss and accelerate insect desiccation (Galovic et al., 2017; doi:10.4154/gc.2017.04).

REPLY: We have incorporated this information in the last sentence of the first paragraph (discussion section). We have also included the following reference Galovic et al., 2017.

  • 49 Galovic I., Halamic J., Grizelj A., Rozman V., Liska A., Korunic Z., Lucic P., Balicevic R. (2017). Croatian diatomites and their possible application as a natural insecticide. Journal of the Croatian Geological Survey and the Croatian Geological Society 70, 27-39.

Reviewer 2 Report

good robust study - maybe reconsider the analytical approach (survival analyses as opposed to ANOVA on percentage values)

good

Author Response

Reviewer 2

Good robust study - maybe reconsider the analytical approach (survival analyses as opposed to ANOVA on percentage values)

REPLY: The authors would like to thank the reviewer for the comments and recommendations on improving the quality of the manuscript before resubmission. A revised version is submitted with the proposed corrections/additions addressed. A point by point reply on each comment/correction can be found below. Before the analysis, the data set were tested for normalization and homogeneity using Levene’s test. However, in some cases, Levene’s test was not equal and O’Brien or Brown-Forsythe tests were utilized. In all cases, the assumptions of ANOVA were met. Thus, we recommend to maintain the statistical analysis as it is provided.

Line 23: You need to say what the secondary species is.

REPLY: We have mentioned the secondary species in the parenthesis. Please see line 36.

Line 120: Too vague – how much?

REPLY: Correct. We have modified this sentence “Lots of soft wheat (500 g/replicate/dose rate) were dusted…”Please see lines 141-142.

Line 121: How was ppm calculated? Parts per million what?

REPLY: ppm was calculated as g/kg of grains, please see line 143.

Line 122: Replicates?

REPLY: It is not referred to replicates. An extra series of soft wheat was left untreated and served as control. Please check the lines 153-158, for the combinations and the replicates that were used in the experimentation.

Line 137: This seems a little strange - in that progeny production is likely affected by 'oviposition substrate' - thus secondary colonisers are unlikely to lay eggs on/near soft wheat?

REPLY: Thank you for this comment. A wide range of stored product beetle species (primary and secondary colonizers) was tested in our bioassays. Both colonizers are able to lay the eggs on the soft wheat. Please check the following references:

  • Rumbos C.I., Pantazis, I., Athanassiou C.G. (2019). Population growth of Aphitobious diaperinus (Coleoptera: Tenebrionidae) on various commodities. Journal of Economic Entomology 113, 1001-1007. https://doi.org/10.1093/jee/toz313
  • Baliota G.V., Lampiri E., Athanassiou C.G. (2022). Different effects of abiotic factors on the insecticidal efficacy of diatomaceous earth against three major stored product beetle species. Agronomy 12, 156. https://doi.org/10.3390/agronomy12010156
  • Athanassiou C.G., Vayias B.J., Dimizas C.B., Kavallieratos N.G., Papagregoriou A.S., Buchelos, C.T., (2005). Insecticidal efficacy of diatomaceous earth against Sitophilus oryzae (L.) (Coleoptera: Curculionidae) and Tribolium confusum Du Val (Coleoptera Tenebrionidae) on stored wheat: Influence of dose rate, temperature and exposure interval. Journal of Stored Products Research 41, 47-55. https://doi.org/10.1016/j.jspr.2003.12.001

Line 139: please comment on why ANOVA was used instead of survival analyses?

percentages generally break the rules of being parametric. I also wonder if the offspring number data would be better analysed using a generalised linear model with a poisson error structure, given it is count data?

REPLY: Please see the reply above.

Line 145: You need to mention MANOVA - this is used in table 1.

REPLY: Revised, see line 169.

Line 352: This is why I believe ANOVA to be the incorrect method of analysis 

REPLY: Thank you for this comment. Indeed, in some cases, low progeny production was observed. However, the assumptions of the analysis of variance have been tested.
